# Evaluation of Environmental *Lactococcus lactis* Strains Reveals Their Potential for Biotransformation of Lignocellulosic Feedstocks

**Desirée Román Naranjo [1,2], Michael Callanan [2] , Anne Thierry [3] and Olivia McAuliffe [1,4,*]**

1 Teagasc Food Research Centre, Moorepark, Fermoy, P61 C996 Cork, Ireland
2 Department of Biological Sciences, Munster Technological University, T12 P928 Cork, Ireland
3 UMR STLO, INRAE, Institut Agro, F-35000 Rennes, France
4 VistaMilk SFI Research Centre, Moorepark, Fermoy, Co., P61 C996 Cork, Ireland
* Correspondence: olivia.mcauliffe@teagasc.ie; Tel.: +353-(0)25-42609

**Abstract:** In this study, the potential for a collection of wild-type *L. lactis* strains to metabolize the breakdown products of lignocellulose was investigated. The strains, isolated from a variety of environmental sources including grass and vegetables, were analyzed for their ability to ferment pentose sugars and their cellulolytic ability. In total, 21 environment-derived *L. lactis* strains were evaluated. Eleven of the 21 *L. lactis* isolates were found to have the potential to ferment pentose sugars commonly produced by lignocellulose breakdown. A 3,5-dinitrosalicylic acid (DNS)-based cellulase assay was performed, and 10 of the 21 *L. lactis* isolates showed cellulolytic activity. Six strains were able to both metabolize pentose sugars and showed cellulolytic activity: these included green pea isolates DPC 6754, DPC 6755, DPC 6756, and DPC 6758, the grass isolate DPC 6760, and the mung bean sprouts isolate KF147. For the first time, certain wild-type non-engineered *L. lactis* were found to possess cellulolytic activity. Moreover, these two abilities do not appear to be correlated. These findings highlight that environment-derived *L. lactis*, a species with a history of safe use in food production, has the potential for second-generation bioconversion processes, and the potential to re-utilize plant biomass found in waste streams.

**Keywords:** lignocellulose; biotransformation; *L. lactis*; environmental isolates; pentoses; cellulolytic; plant biomass; waste streams

## 1. Introduction

As climate change records indicate, global $CO_2$ emissions derived from fossil fuels have risen for three consecutive years, with all economic indicators suggesting this growing trend to continue in the future [1]. As the Sixth Assessment Report of the Intergovernmental Panel on Climate Change (IPCC) states, the global surface temperature will continue to increase, exceeding 2 °C, unless deep reductions in $CO_2$ levels and other greenhouse gas emissions take place [2]. Moreover, the energy demand is constantly increasing, with estimates of a 28% increase of total energy consumption worldwide by 2040 [3]. The deteriorating climate change situation and increasing energy demands, together with the depletion of fossil fuels, makes the search for other renewable sources for fuel production an urgent one. Biofuel, derived from the conversion of biomass, is one of the most promising renewable alternatives to fossil fuels [4].

Lignocellulose, a polymer that is the most abundant biomaterial on Earth and one of the most abundant renewable feedstocks, comprises on average 23% lignin, 40% cellulose, and 33% hemicellulose by dry weight [5,6]. Lignocellulose is the main component of forestry, agricultural, and agro-industrial wastes. This agro-industrial waste source has been reported as having potential as an inexpensive, abundant, and renewable energy source. Lignocellulose biomass can be hydrolyzed using physical, chemical, and biological

methods. Biological treatments, such as microbial biotransformation, offer several advantages including high specificity, low energy consumption, no chemical requirement, and mild environmental conditions avoiding sugar degradation [7]. Nonetheless, biological treatments also have some disadvantages such as a low hydrolysis rate [7].

Lignocellulose is composed of closely associated polymers such as cellulose, hemicellulose, and lignin, which form a complex cellular matrix in the vegetal biomass. However, the amount of each polymer is different between the plant species [7,8]. Cellulose can be converted into fermentable sugars with an appropriate cellulolytic system involving the production of cellulase enzyme systems by many cellulolytic bacterial genera such as *Clostridium*, *Cellulomonas*, *Bacillus*, and *Actinomycetes* [9,10]. These cellulolytic systems have been applied to many industrial processes including biofuel production, or agricultural and plant waste management [10]. Cellulolytic enzymatic systems comprise three main cellulase groups: endoglucanases, exoglucanases, and β-glucosidases, depending on where in the molecule the cleavage takes place [11]. Hemicelluloses are a family of complex biopolymers that can be degraded by specific hemicellulase-producing microorganisms. These hemicellulases include the enzymes xylanase and manannase, produced mostly by fungal species due to their complex natures. Lignin is the second most abundant renewable polymer after cellulose. It is also the most structurally complex lignocellulose component (highly cross-linked phenylpropanoid units), which makes it resistant to its use by microorganisms to produce monomers [12,13].

Previous studies have found that certain lactic acid bacteria species possess the capability to ferment several cheap waste streams whose main component is lignocellulose, resulting in the generation value-added products, primarily lactic acid, but also ethanol, butanol, or polylactic acid. *Pediococcus acidilactici* DSM 20284 and *Pediococcus pentosaceus* ATCC 25745 were found to ferment common pentose sugars found in lignocellulose, such as xylose and arabinose [14]. Other examples include the production of L-lactic acid from sugarcane bagasse by *Lactiplantibacillus delbrueckii* NCIM 2025 [15], from wheat straw hemicellulose hydrolysate by *Lactiplantibacillus pentosus* CHCC 2355 and *Levilactobacillus brevis* CHCC 2097 [16], or from lignocellulosic hydrolysates by *Lactiplantibacillus* sp. RKY2 KCTC 10353BP [17]. Moreover, the co-cultures of *Levilactobacillus brevis* and *Lactiplantibacillus plantarum* were able to produce lactic acid from biomass-derived sugars co-fermentation [18].

It has previously been shown that *Lactococcus lactis* strains isolated from environmental sources, including plant material, have more diverse metabolic capabilities compared to their dairy counterparts, in terms of lipolytic activity [19] or volatile compound production [20,21]. To date, studies on the lignocellulose metabolization capabilities of *L. lactis* have mostly focused on the ability of genetically engineered *L. lactis* strains to ferment lignocellulose [22–24]. However, very few studies focus on the natural ability of *L. lactis* environmental isolates to biotransform this complex carbon source, and those studies only focus on the strain *L. lactis* IO-1, which was found to be capable of fermenting xylose [25,26]. Even though the mutant *L. lactis* strain IL 1403 produced 50.1 g $L^{-1}$ of L-lactic acid from xylose [22], the wild strain *L. lactis* IO-1 produced a maximum concentration of 33.26 g $L^{-1}$ of lactic acid from xylose [27]; this shows the potential of non-engineered *L. lactis* to metabolize this complex carbon source and to generate compounds of interest. Furthermore, most studies focus on the presence of genes related to these activities from a genomic perspective [28,29]. The main difference between the physiological and the genomic perspective is the lack of correlation between the presence of specific genes and the phenotypic activity assigned to those genes, which happens sometimes, mainly due to gene inactivation [28]. For example, the presence of a xylose utilization gene cluster in strain *L. lactis* subsp. *lactis* IL-1403 did not correlate with the ability of this strain to utilize xylose. In this study, the objective was to evaluate a collection of *L. lactis* strains derived from grass, vegetable, and other environmental sources for potential lignocellulose biotransformation from a physiological point of view.

## 2. Materials and Methods

### 2.1. Bacterial Strains and Culture Conditions

Details of the strains used in this study are listed in Table 1. *Pediococcus acidilactici* DSM 20284 was purchased from DSMZ and was used as a positive control for the pentose utilization test. The strains are maintained in the DPC Culture Collection (Teagasc Food Research Centre, Moorepark, Ireland) or were purchased from the German Collection of Microorganisms and Cell Cultures (DSMZ, Braunschweig, Germany). *L. lactis* strains were cultured in M17 media (Oxoid, Hamphsire, England) supplemented with 5 g $L^{-1}$ D (+)-glucose monohydrate (G-M17) (VWR, Leuven, Belgium). *Lactiplantibacillus* and *Levilactobacillus* strains were grown in MRS medium (Oxoid, Hamphsire, England). All the strains were incubated aerobically at 30 °C.

**Table 1.** Bacterial strains used in this study.

| Strain Number | Species/Subspecies (ssp) | Isolation Source | Source or Reference |
|---|---|---|---|
| DSM 20284 | *Pediococcus acidilactici* | Barley | [30] |
| KF147 | *Lactococcus lactis* | Mung bean sprouts | [31], DPC CC [1] |
| DPC 6754 | *Lactococcus lactis* ssp. *lactis* | Green peas | [20], DPC CC |
| DPC 6755 | *Lactococcus lactis* ssp. *lactis* | Green peas | [20], DPC CC |
| DPC 6756 | *Lactococcus lactis* ssp. *lactis* | Green peas | [20], DPC CC |
| DPC 6757 | *Lactococcus lactis* ssp. *lactis* | Green peas | [20], DPC CC |
| DPC 6758 | *Lactococcus lactis* ssp. *lactis* | Green peas | [20], DPC CC |
| DPC 6759 | *Lactococcus lactis* ssp. *lactis* | Green peas | [20], DPC CC |
| DPC 6760 | *Lactococcus lactis* ssp. *lactis* | Grass | [20], DPC CC |
| DPC 6761 | *Lactococcus lactis* ssp. *lactis* | Grass | [20], DPC CC |
| DPC 6762 | *Lactococcus lactis* ssp. *lactis* | Baby corn | [20], DPC CC |
| DPC 6763 | *Lactococcus lactis* ssp. *lactis* | Grass | [20], DPC CC |
| DPC 6764 | *Lactococcus lactis* ssp. *cremoris* | Grass | [20], DPC CC |
| DPC 6765 | *Lactococcus lactis* ssp. *cremoris* | Grass | [20], DPC CC |
| DPC 6853 | *Lactococcus lactis* ssp. *lactis* | Corn | [21], DPC CC |
| DPC 6854 | *Lactococcus lactis* ssp. *cremoris* | Grass | [21], DPC CC |
| DPC 6855 | *Lactococcus lactis* ssp. *cremoris* | Grass | [21], DPC CC |
| DPC 6856 | *Lactococcus lactis* ssp. *cremoris* | Rumen | [21], DPC CC |
| DPC 6857 | *Lactococcus lactis* ssp. *cremoris* | Grass | [21], DPC CC |
| DPC 6858 | *Lactococcus lactis* ssp. *cremoris* | Grass | [21], DPC CC |
| DPC 6859 | *Lactococcus lactis* ssp. *cremoris* | Grass | [21], DPC CC |
| DPC 6860 | *Lactococcus lactis* ssp. *cremoris* | Grass | [21], DPC CC |
| DPC 6417 | *Lactiplantibacillus plantarum* | Teat rinse | DPC CC |
| DPC 6421 | *Lactiplantibacillus plantarum* | Bovine feces | DPC CC |
| DPC 6427 | *Levilactobacillus brevis* | Silage | DPC CC |
| DPC 6428 | *Lactiplantibacillus plantarum* | Silage | DPC CC |
| DPC 6429 | *Lactiplantibacillus plantarum* | Hand rinse | DPC CC |
| DPC 6430 | *Lactiplantibacillus plantarum* | Water milking yard | DPC CC |
| DPC 6606 | *Lactiplantibacillus pentosus* | Olives | DPC CC |
| DPC 6607 | *Lactiplantibacillus pentosus* | Olives | DPC CC |
| DPC 6608 | *Lactiplantibacillus pentosus* | Olives | DPC CC |
| DPC 6609 | *Lactiplantibacillus pentosus* | Olives | DPC CC |
| DPC 6616 | *Lactiplantibacillus pentosus* | Silage | DPC CC |
| DPC 6617 | *Lactiplantibacillus pentosus* | Silage | DPC CC |
| DPC 6618 | *Lactiplantibacillus pentosus* | Silage | DPC CC |
| DPC 6619 | *Lactiplantibacillus pentosus* | Silage | DPC CC |
| DPC 6620 | *Lactiplantibacillus pentosus* | Olives | DPC CC |
| DPC 6622 | *Lactiplantibacillus pentosus* | Olives | DPC CC |
| DPC 6623 | *Lactiplantibacillus pentosus* | Olives | DPC CC |
| DPC 6624 | *Lactiplantibacillus pentosus* | Olives | DPC CC |
| DPC 6625 | *Lactiplantibacillus pentosus* | Olives | DPC CC |
| DPC 6627 | *Lactiplantibacillus pentosus* | Olives | DPC CC |
| DPC 6628 | *Lactiplantibacillus pentosus* | Olives | DPC CC |
| DPC 6667 | *Lactiplantibacillus plantarum* | Silage | DPC CC |
| DPC 6672 | *Lactiplantibacillus plantarum* | Silage | DPC CC |
| DPC 6682 | *Lactiplantibacillus plantarum* | Japanese pickles | DPC CC |
| DPC 6730 | *Levilactobacillus brevis* | Japanese pickles | DPC CC |
| DPC 7020 | *Lactiplantibacillus plantarum* | Rumen | DPC CC |

[1] DPC CC is the Teagasc DPC Culture Collection housed at the Teagasc Food Research Centre, Moorepark, Fermoy, Cork, Ireland.

*2.2. Pentose Utilization Agar Test*

The preliminary qualitative ability to utilize xylose and arabinose was tested through spotting onto modified 10% MRS agar medium (mMRS). The basal composition of the 10% mMRS agar medium was (per liter) (all supplied by Merck): 1 g casein peptone (tryptic digest), 15 g agar, 0.8 g meat extract, 0.4 g yeast extract, 1 mL Tween 80, 2 g $K_2HPO_4$, 5 g sodium acetate, 2 g diammonium citrate, 0.2 g $MgSO_4 \cdot 7H_2O$ and 0.05 g $MnSO_4 \cdot H_2O$; and 20 g of carbon source: glucose or galactose (hexoses), and xylose or arabinose (pentoses), as described by Boguta et al., 2014 [14]. A 5 µL aliquot of an overnight culture in the appropriate broth was spotted onto the agar plates. Following incubation at 30 °C for 48 h, the growth of the strains was evaluated as good growth (+), moderate (/), and no growth (−), using as a reference the growth of the same strain on 10% mMRS + glucose agar plates. All of the strains were tested in triplicate for each sugar.

*2.3. Pentose Utilization Growth Curves*

Strains that were positive in the pentose utilization agar test (for both pentose sugars) were further tested in modified 10% mMRS broth + 20 g of carbon source: glucose or galactose (hexoses), and xylose or arabinose (pentoses). The media composition was as described above, but excluding the agar. The assay was performed in a transparent 96-well plate (Sarstedt Ltd., Leicester, UK). The reaction mixture included 200 µL of 10% modified mMRS + glucose, galactose, arabinose or xylose, and 2 µL of overnight culture of the selected strains. The mixture was incubated at 30 °C for 48 h, and the absorbance at 600 nm ($A_{600}$) was measured with Epoch 2 Microplate Spectrophotometer (BioTek, Winooski, VT, USA) every hour during the 48 h. All the assays were performed in triplicate.

In order to compare the growth between the different strains and sugars, the growth rates were calculated for each strain/sugar combination. Raw data were gathered in Microsoft Office Excel 2013 and the statistical software R 4.0.1. The growthcurver package was used to fit the growth data [32,33]. The growth rate was calculated with the Spline model for the best performing strains, with inclusion of the lag (λ) parameter, meaning that the exponential growth of the biomass of the bacteria is only calculated at a given OD.

*2.4. Qualitative Cellulolytic Screening*

Carboxymethyl cellulose agar (CMC agar) was used to identify the presence or absence of endo-cellulolytic activity. The CMC agar composition was as follows: 11.28 g $L^{-1}$ M9 minimal medium salts (MP Biomedicals, Eschwege, Germany), 5 g $L^{-1}$ carboxymethyl cellulose (Merck, Arklow, Ireland), and 15 g $L^{-1}$ agar (Merck). A 5 µL aliquot of 2% (*v/v*) of an overnight culture was spotted onto the CMC agar and incubated for 2 days at 30 °C. Following incubation, the agar plates were flooded with Gram's iodine solution (Merck) for 3 min to visualize the cellulolytic activity [11].

*2.5. Cellulolytic Activity Assay*

Reducing sugars were measured by following a modified 3,5-dinitrosalicylic acid (DNS) assay as described in [34].

Cell-free extracts (CFE) containing crude enzymes were prepared fresh prior to use on the same day of the assays. A total of 2% (*v/v*) of a 24 h culture (with an average $OD_{600} = 0.812$) was inoculated in 10 mL of the corresponding broth for each bacterial strain and incubated for 24 h at 30 °C. In order to obtain the cell-free-extract (CFE) containing crude enzymes, cells were centrifuged at $10,000\times g$ for 20 min at 4 °C and kept on ice until further use.

The reaction mixture consisted of 100 µL of 1% (*w/v*) carboxymethyl cellulose (CMC, Merck) dissolved in 0.05 mol $L^{-1}$ glycine/NaOH buffer (pH = 9, Merck) (substrate of the assay) and 20 µL of CFE in a final volume of 120 µL. The reaction mixture was incubated for 30 min at 37 °C, and the enzymatic reaction was halted by placing the reactions on ice. A total of 360 µL of DNS solution was added to the reaction mixture, which was then placed in a boiling water bath for 5 min and subsequently placed on ice to stop the reaction.

Samples were then diluted appropriately with sterile water (1:2), vortexed, and 200 μL was pipetted into a 96-well plate. Absorbance was read at 550 nm wavelength using an Epoch 2 Microplate spectrophotometer (Agilent BioTek, Winooski, VT, USA) in triplicates.

The DNS solution was prepared as follows: 3,5-dinitrosalicylic acid (10 g L$^{-1}$, Merck), NaOH (20 g L$^{-1}$, Merck), sodium sulfite anhydrous (0.5 g L$^{-1}$, Fisher Scientific, Dublin, Ireland), potassium sodium tartrate tetrahydrate (200 g L$^{-1}$, Merck), and phenol (2 g L$^{-1}$, Merck). The 3,5-dinitrosalicylic acid and NaOH solution were dissolved separately and added to the mixed solution of sodium sulfite anhydrous and potassium sodium tartrate tetrahydrate, and finally, phenol was added to the solution.

A glucose standard curve was generated to quantify the amount of reducing sugar in the reaction mixture. One cellulolytic unit was defined as the enzymatic activity that liberates one microgram of glucose per min per mL CFS. A stock solution of glucose (3 mg mL$^{-1}$, Merck) was used to prepare the glucose solutions with different concentrations ranging from 0 to 0.9 mg mL$^{-1}$.

## 3. Results

### 3.1. L. lactis Strains Derived from Green Peas, Grass, and Corn Can Metabolize Arabinose and Xylose

Results for all the strains tested are included in Tables S1 and S2. A preliminary qualitative test was performed on the *L. lactis* strain collection to evaluate the ability of the environmental *L. lactis* strains to utilize the pentose sugars xylose and arabinose. Overnight cultures of each strain were spotted onto modified 10% mMRS agar medium containing the sugar of interest, and bacterial growth was indicated as good (+) where clear visible bacterial growth was observed, moderate (/) where faint bacterial growth was observed, and no growth (−) where no growth was observed, in comparison with 10% mMRS + glucose. *P. acidilactici* DSM 20284 served as the positive control, as this strain has previously been shown to utilize the pentose sugars under investigation in this study (Boguta et al., 2014). Examples of the plate assays can be observed in Figure 1.

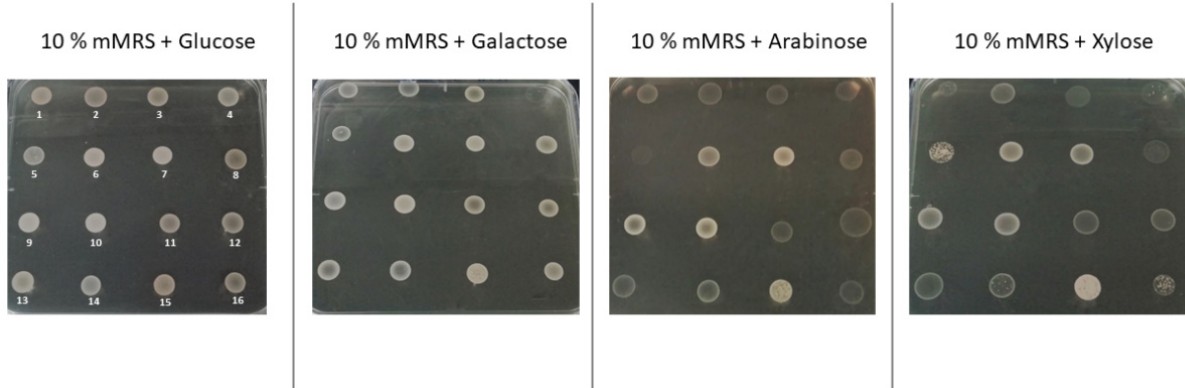

**Figure 1.** Pentose utilization agar test, with 16 strains (out of the 47 strains tested) after 48 h of incubation at 30 °C. The same strains across all plates are: (1) *Lactiplantibacillus pentosus* DPC 6627, (2) *P. acidilactici,* (3) *Lactococcus lactis* ssp. *lactis* DPC 6757, (4) *Lactiplantibacillus pentosus* DPC 6607, (5) *Lactiplantibacillus pentosus* DPC 6624, (6) *Lactiplantibacillus pentosus* DPC 6619, (7) *Lactiplantibacillus pentosus* DPC 6620, (8) *Lactiplantibacillus pentosus* DPC 6618, (9) *Lactiplantibacillus pentosus* DPC 6622, (10) *Lactiplantibacillus pentosus* DPC 6623, (11) *Lactiplantibacillus plantarum* DPC 7020, (12) *Lactiplantibacillus plantarum* DPC 6667, (13) *Lactococcus lactis* ssp. *lactis* DPC 6758, (14) *Lactococcus lactis* ssp. *lactis* DPC 6759, (15) *Lactococcus lactis* ssp. *lactis* DPC 6760, and (16) *Lactiplantibacillus pentosus* DPC 6607.

While all of the 21 environmental *L. lactis* strains tested were capable of growth on the hexose sugars glucose and galactose, only 11 strains exhibited growth in the presence of the pentose sugars arabinose and xylose as the sole sugar source. These included strains derived from green peas (DPC 6754, DPC 6755, DPC 6756, DPC 6758, and DPC 6759),

grass (DPC 6760, DPC 6763, DPC 6764, and DPC 6765), baby corn (DPC 6762), and mung bean sprouts (KF147). For comparison, we included a number of species in our assay that had previously been shown to have the potential to metabolize lignocellulose. These included *Lactiplantibacillus pentosus, Levilactobacillus brevis,* and *Lactiplantibacillus plantarum* (Garde et al., 2002; Zhang and Vadlani, 2015). Of the 15 *L. pentosus* strains tested, five strains—DPC 6616, DPC 6619, and DPC 6620, isolated from silage; and DPC 6622 and DPC 6623, isolated from olives—were also able to grow on arabinose and xylose. The two *L. brevis* strains tested—DPC 6427, isolated from silage, and DPC 6730, isolated from Japanese pickles—were able to grow on arabinose and xylose. Of the nine *L. plantarum* strains tested, only one strain—DPC 6429, isolated from hand rinse—was able to grow on arabinose and xylose.

The 11 strains that were determined to exhibit good growth (+) in the qualitative agar-based assay were further tested in liquid media, and growth rates were calculated. Growth rates were highly variable, so mean growth rates were calculated using the Spline growth model (Table 2).

**Table 2.** Growth rates of test strains on hexoses and pentoses (Spline model). The mean of the maximum growth rate ($OD_{600}$ $\mu_{max}$ ($h^{-1}$)) and standard deviation (SD) for every strain/sugar combination is given.

| Mean Growth Rates ($OD_{600}$ $\mu_{max}$ ($h^{-1}$)) | | | | | | | | | |
| --- | --- | --- | --- | --- | --- | --- | --- | --- | --- |
| | | Glucose | | Galactose | | Arabinose | | Xylose | |
| DPC Code [1] | Species/Subspecies (ssp) | Mean | SD | Mean | SD | Mean | SD | Mean | SD |
| - | *P. acidilactici* | 0.181 | 0.003 | 0.129 | 0.027 | 0.086 | 0.003 | 0.062 | 0.007 |
| KF147 | *Lactococcus lactis* ssp. *lactis* | 0.075 | 0.002 | 0.064 | 0.005 | 0.051 | 0.001 | 0.031 | 0.003 |
| 6754 | *Lactococcus lactis* ssp. *lactis* | 0.160 | 0.003 | 0.062 | 0.004 | 0.053 | 0.006 | 0.034 | 0.004 |
| 6755 | *Lactococcus lactis* ssp. *lactis* | 0.159 | 0.001 | 0.050 | 0.002 | 0.051 | 0.005 | 0.035 | 0.008 |
| 6756 | *Lactococcus lactis* ssp. *lactis* | 0.089 | 0.004 | 0.044 | 0.003 | 0.048 | 0.005 | 0.026 | 0.007 |
| 6758 | *Lactococcus lactis* ssp. *lactis* | 0.174 | 0.003 | 0.074 | 0.003 | 0.081 | 0.002 | 0.035 | 0.010 |
| 6759 | *Lactococcus lactis* ssp. *lactis* | 0.103 | 0.008 | 0.039 | 0.006 | 0.018 | 0.004 | 0.018 | 0.002 |
| 6760 | *Lactococcus lactis* ssp. *lactis* | 0.085 | 0.013 | 0.130 | 0.012 | 0.064 | 0.003 | 0.033 | 0.007 |
| 6762 | *Lactococcus lactis* ssp. *lactis* | 0.091 | 0.007 | 0.027 | 0.009 | 0.078 | 0.003 | 0.029 | 0.008 |
| 6763 | *Lactococcus lactis* ssp. *lactis* | 0.170 | 0.009 | 0.111 | 0.005 | 0.056 | 0.006 | 0.035 | 0.004 |
| 6764 | *Lactococcus lactis* ssp. *cremoris* | 0.180 | 0.003 | 0.072 | 0.007 | 0.076 | 0.004 | 0.046 | 0.003 |
| 6765 | *Lactococcus lactis* ssp. *cremoris* | 0.096 | 0.004 | 0.051 | 0.003 | 0.014 | 0.003 | 0.018 | 0.003 |
| 6427 | *Levilactobacillus brevis* | 0.041 | 0.006 | 0.050 | 0.003 | 0.077 | 0.004 | 0.077 | 0.019 |
| 6429 | *Lactiplantibacillus plantarum* | 0.115 | 0.020 | 0.052 | 0.009 | 0.055 | 0.003 | 0.037 | 0.005 |
| 6616 | *Lactiplantibacillus pentosus* | 0.137 | 0.009 | 0.046 | 0.003 | 0.049 | 0.009 | 0.020 | 0.004 |
| 6619 | *Lactiplantibacillus pentosus* | 0.217 | 0.004 | 0.139 | 0.020 | 0.095 | 0.003 | 0.064 | 0.012 |
| 6620 | *Lactiplantibacillus pentosus* | 0.209 | 0.013 | 0.133 | 0.009 | 0.085 | 0.009 | 0.043 | 0.004 |
| 6622 | *Lactiplantibacillus pentosus* | 0.223 | 0.008 | 0.135 | 0.015 | 0.097 | 0.008 | 0.060 | 0.005 |
| 6623 | *Lactiplantibacillus pentosus* | 0.211 | 0.020 | 0.128 | 0.008 | 0.089 | 0.011 | 0.056 | 0.007 |
| 6730 | *Levilactobacillus brevis* | 0.071 | 0.005 | 0.017 | 0.003 | 0.018 | 0.004 | 0.028 | 0.003 |

[1] DPC CC is the Teagasc DPC Culture Collection housed at the Teagasc Food Research Centre, Moorepark, Fermoy, Cork, Ireland.

Generally, most of the environmental *L. lactis* tested shown in Table 2 were able to ferment arabinose and xylose as the sole carbon source, with comparable $\mu_{max}$ values by *Lactiplantibacillus pentosus* and *Levilactobacillus brevis* strains, which were previously shown to grow in these sugars. Most strains had higher growth rates on arabinose rather than xylose, and only two strains had higher growth rates in xylose compared to arabinose: DPC6765 and DPC6730. *L. lactis* strains DPC6758 and DPC6762, isolated from green peas and baby corn, had comparable growth rates of 0.081 and 0.078 growing in arabinose, respectively. These growth rates are comparable to the four strains with the highest $\mu_{max}$ (0.085–0.097) growing in arabinose, which were all *Lactiplantibacillus pentosus* strains (DPC 6622, DPC 6619, DPC 6623, and DPC 6620). With regards to xylose, the four strains with the highest $\mu_{max}$ (0.056–0.077) were *Levilactobacillus brevis* strain DPC 6427 and *Lactiplantibacillus*

*pentosus* strains DPC 6619, DPC 6622, and DPC 6623. The environmental *L. lactis* strain DPC 6764 showed with a comparable growth rate of 0.046, and all the rest of the environmental *L. lactis* strains had $\mu_{max}$ values ranging from 0.018 to 0.035. It was of interest that the best performing strains could grow as well on arabinose and xylose as galactose, which is released during the fermentation of the major milk sugar, lactose.

### 3.2. Environmental L. lactis Strains Exhibit Cellulolytic Activity

The preliminary qualitative cellulolytic assay (CMC agar) allowed for the identification of putative positive strains based on the appearance of a halo on the cell culture after flooding the agar plate with Gram's iodine (Figure 2).

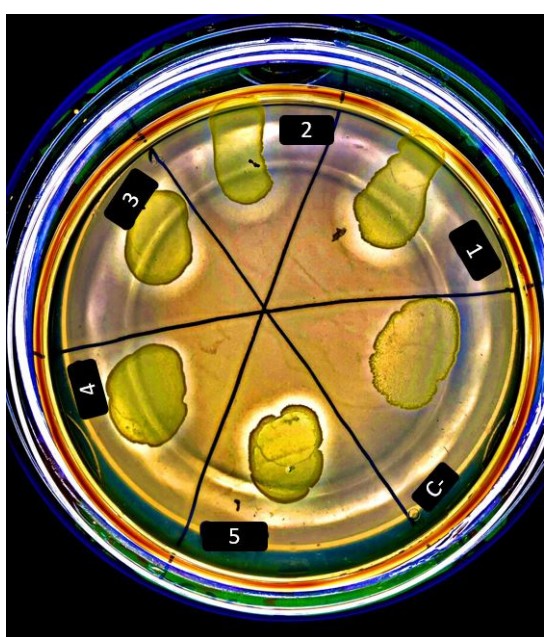

**Figure 2.** CMC agar test showing 5 cellulolytic positive strains (halo zone after flooding with Gram's iodine) and a negative strain (C-). The appearance of a halo on the cell culture indicates cellulose metabolization in the positive strains. Strains shown are: (1) *L. lactis* ssp. *lactis* KF147, (2) *L. lactis* ssp. *lactis* DPC 6760, (3) *L. lactis* ssp. *lactis* DPC 6761, (4) *L. lactis* ssp. *lactis* DPC 6754, (5) *L. lactis* ssp. *lactis* DPC 6755, and (C-) *L. brevis* DPC 6427.

The strains that showed the halo were classified as positive (+) strains (Figure 3). All the positive strains are *L. lactis*, isolated from environmental sources such as green peas and grass. None of the other species tested (*Lactiplantibacillus pentosus*, *Levilactobacillus brevis* and *Lactiplantibacillus plantarum*) showed cellulolytic activity. Strains that tested positive on the agar assay were selected for further quantification of cellulolytic activity degradation (DNS-based enzymatic assay), by calculating the cellulolytic activity (mg mL$^{-1}$ min$^{-1}$).

As reported in Figure 3, all 10 strains are *L. lactis* environmental isolates, of the 21 environmental isolates tested. These positive strains include five isolates derived from grass, (DPC 6760, DPC 6761, DPC 6855, DPC 6858, and DPC 6860), four isolates derived from green peas, (DPC 6754, DPC 6755, DPC 6756, and DPC 6758), and one isolate derived from mung bean sprouts, KF147. Those strains were further tested in a quantitative assay. The strains with the highest cellulolytic activity (mg mL$^{-1}$ min$^{-1}$) were DPC 6761 (*Lactococcus lactis* isolated from grass), and DPC 6760 (*Lactococcus lactis* isolated from grass), with a value of 0.839 and 0.755 mg mL$^{-1}$ min$^{-1}$, respectively (Figure 3). This proves the previously unexplored capability of *L. lactis* environmental isolates to metabolize cellulose and potentially contribute to lignocellulose degradation.

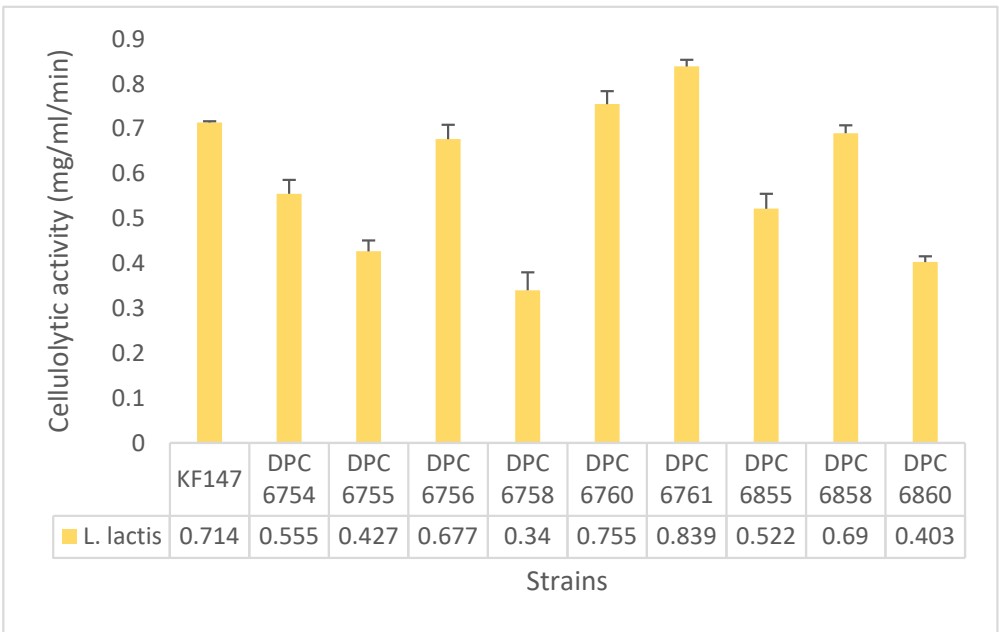

**Figure 3.** Cellulolytic activity (mg mL$^{-1}$ min$^{-1}$) in the quantitative DNS assay for all the environmental *L. lactis* isolates that were positive on CMC agar. The values represent the average of triplicates (mean), the error bars, and the standard deviation (SD).

In order to facilitate the visualization of all the strains used in this study and all the phenotypic tests performed, a Supplementary Table is provided (Table S3). Overall, only six environmental *L. lactis* strains were able to both metabolize pentoses and showed cellulolytic activity—green peas isolates DPC 6754, DCP 6755, DPC 6756, and DPC 6758, grass isolate DPC 6760, and mung bean sprouts isolate KF147 (Table S3). Some strains that were able to metabolize arabinose and xylose did not show cellulolytic activity—DPC 6759, isolated from green peas, DPC 6762, isolated from baby corn and DPC 6763, DPC 6764, and DPC 6765, isolated from grass. On the other hand, some *L. lactis* environmental isolates with cellulolytic activity could not metabolize arabinose and xylose—DPC 6761, DPC 6855, DPC 6858, and DPC 6860—all isolated from grass. This shows that these two abilities (cellulolytic activity and pentose sugars fermentation) do not appear to be correlated.

## 4. Discussion

In this study, the ability of six *L. lactis* strains isolated from diverse, primarily plant-based environments to potentially metabolize plant biomass (whose main component is lignocellulose) was demonstrated, by proving their ability to metabolize pentoses commonly derived from lignocellulose (arabinose and xylose), and by showing cellulolytic activity. Initially, their ability to grow on the main fermentable pentoses released after lignocellulose breakdown (arabinose and xylose) was demonstrated. Out of the 21 *L. lactis* strains, 11 were able to grow in the presence of arabinose and xylose as the sole sugar source. Those strains were isolated from green peas, grass, baby corn, and mung bean sprouts. *L. lactis* DPC 6758 and DPC 6762, isolated from green peas and baby corn, had comparable growth rates to the highest $\mu_{max}$ values *of Lactiplantibacillus pentosus* and *Levilactobacillus brevis* strains growing in arabinose. Moreover, environmental *L. lactis* strains also showed $\mu_{max}$ values comparable to the highest values of *Levilactobacillus brevis and Lactiplantibacillus pentosus* strains growing in xylose as the sole sugar source. This proves the ability of certain environmentally derived *L. lactis* strains to metabolize the pentose sugars released after lignocellulose breakdown: arabinose and xylose. Ten *L. lactis* strains out of the 21 tested showed cellulolytic activity in the qualitative agar-based assay and were further tested to determine the extent of cellulolytic activity. Out of the 21 strains, six environmental *L.*

*lactis* strains were able to both metabolize the pentose sugars and had cellulolytic activity, proving our hypothesis about their potential to metabolize lignocellulosic feedstocks.

The diverse metabolic capabilities and lifestyle adaptations of wild type 'non-dairy' (here referred to as 'environmental' [35]) *L. lactis* strains have previously been shown. One of those capabilities is related to their ability to metabolize a wide variety of sugars. Previous studies have revealed how some plant isolates (in contrast to their dairy counterparts) were able to grow on sugars like arabinose and xylose (Siezen et al., 2008; Passerini et al., 2013). One of these strains was *L. lactis* KF147, used in this study as a positive control, which proved the same capabilities to metabolize those pentose sugars. However, very few other environmental *L. lactis* strains had been tested on those pentoses sugars to prove this theory. This study supports that hypothesis. Moreover, previous studies have shown the presence of genes involved in the breakdown of plant polymers in a diverse group of environmental *L. lactis* strains through comparative genome hybridization [29]. These genes include those involved in the arabinose metabolism, such as the complete arabinose gene cluster araADBTFPR, and xylose metabolism, such as genes xynA, xynP, xynQ, and xynS. A preliminary analysis of the available genomes from our strain collection (*L. lactis* ssp. *lactis* KF147, *L. lactis* ssp. *lactis* DPC 6756, *L. lactis* ssp. *lactis* DPC 6853, *L. lactis* ssp. *cremoris* DPC 6856, *L. lactis* ssp. *cremoris* DPC 6855, and *L. lactis* ssp. *cremoris* DPC 6860) using Scoary [36] did not clearly show a genotype–phenotype correlation with regard to the metabolism of these sugars. Further whole genome sequencing is underway in order to understand the genetic basis for the sugar metabolism demonstrated here.

This study proves the diverse capabilities of certain environmental *L. lactis* strains to ferment plant-related sugars such as arabinose and xylose. Apart from more diversity in sugar metabolization, other diverse capabilities have been reported in environmental *L. lactis* isolates, such as the presence of gene clusters for exopolysaccharide production [28], and the esterolytic activity of the environmental strains [19]. In terms of cellulolytic activity, little is known about the ability of wild-type *L. lactis* to metabolize cellulose. Previous studies have constructed engineered *L. lactis* strains with genes coding for a GH6 cellulase, endoglucanase, cellobiohydrolase, and β-glucosidase, in order to be able to metabolize cellulose into lactic acid [24,37,38]. In this study, the untapped cellulolytic activity of wild environmental *L. lactis* was revealed. In fact, no native cellulolytic *L. lactis* has ever been identified before [39].

One of the main challenges of lignocellulosic feedstocks metabolization via LAB is the need for physical, and/or chemical, and/or enzymatic pre-treatment(s) of biomass, which in turn releases inhibitory compounds to the media, as well as an inefficient saccharification of biomass [39]. The resistance to such inhibitory compounds should be tested in future studies with those strains that were both able to metabolize cellulose and pentoses sugars. In the past, LAB have been co-cultivated with native cellulolytic microorganisms in order to replace the saccharification of lignocellulosic biomass via exogenously supplemented cellulases. Several studies have successfully converted cellulolytic biomass to products like butanol or ethanol using this co-cultivation approach [40,41]. These limitations have added to the high cost of the added cellulases [42], and could be overcome with the use of native cellulolytic LAB strains, like the ones reported in this study. However, further phenotypic research is needed in order to evaluate the capability of those environmental *L. lactis* strains screened in a real-life plant biomass derived from a plant-based waste stream. Even though the most well-studied commercial product for the *Lactiplantibacillus* genus from lignocellulose bioconversion is lactic acid [43], other industrially relevant compounds, such as biofuel, could be also explored [44].

The *L. lactis* strains used in this study were compared to the species *Lactiplantibacillus pentosus*, *Lactiplantibacillus plantarum*, and *Levilactobacillus brevis*. *Lactiplantibacillus* is the widest and most diverse genus of lactic acid bacteria [45]. Even though LAB are commonly found in dairy environments, previous research has shown the niche adaptation of lactobacilli strains isolated from other environments, such as meat, plant, and sourdough [46]. In this study, *L. lactis* strains did not only show their ability to metabolize

specific pentose sugars, but also the growth rates in the pentoses arabinose and xylose were very comparable to the best-performing lactobacilli. The high performances of other species such as *Lactiplantibacillus pentosus*, *Lactiplantibacillus plantarum*, and *Levilactobacillus brevis* was expected, since they had previously shown their ability to utilize the pentose sugars [14,16,18]. However, only some of the *L. lactis* strains showed cellulolytic activity, and none of the other three species evaluated in this study (in the preliminary qualitative assay) showed cellulolytic activity. This is interesting, since even though most studies rely on the addition of external cellulolytic enzymes for the bioconversion of lignocellulosic material by *L. pentosus* and *L. plantarum* [47,48], some recent studies have found strains like *L. plantarum* RI 11 with the ability to produce extracellular cellulolytic enzymes, such as endoglucanase, exoglucanase, and β-glucosidase, without the addition of external enzymes [49]. However, the *L. plantarum* screened in this study were not able to grow on cellulose. Nonetheless, we understand recent concerns about the use and the accuracy of a DNS assay for the reducing sugars, since other carbonyl groups could potentially react with the DNS, leading to incorrect yields of reducing sugars [50]. These results are in accordance to those found in previous studies, where it was found that even though *L. plantarum* contains 55 genes encoding 18 glycoside hydrolases, none of them are strict cellulases [51]. Therefore, *L. plantarum* strains may lack the inherent ability to metabolize cellulose, while some *L. lactis* strains tested in this study may have acquired the ability to metabolize cellulose in response to niche specialization.

These findings highlight that wild environmental *Lactococcus lactis* have the potential to become platforms for second-generation bioconversion processes. Further work is required in order to test the biotransformation potential of these strains in a cellulosic waste stream substrate. The significance of this study relies on the possibility for potentially using these environmental strains to biotransform plant biomass into products such as bio-fuel and organic chemicals [52], helping to cope with the increasing energy demand, increasing temperatures, and becoming a renewable alternative to fossil fuels.

**Supplementary Materials:** The following supporting information can be downloaded at: https://www.mdpi.com/article/10.3390/applmicrobiol2040061/s1. Table S1: Bacterial growth on the four different sugars, indicated as good (+) where clear visible bacterial growth was observed, moderate (/) where faint bacterial growth was observed, and no growth (−) where no growth was observed, in comparison with 10% mMRS + glucose, for the Study group: non-engineered environmental *L. lactis* strains used in this study. Table S2: Bacterial growth on the four different sugars, indicated as good (+) where clear visible bacterial growth was observed, moderate (/) where faint bacterial growth was observed, and no growth (−) where no growth was observed, in comparison with 10% mMRS + glucose for other species/subspecies with proven lignocellulose metabolization potential used in this study for comparison to our study group. Table S3: All the bacterial strains used in this study, including their isolation source and the main tests performed in this study. Pentose utilization test refers to (+) when the strain was able to grow in all four sugars (glucose, galactose, arabinose, and xylose). mMRS + Glucose, mMRS + Galactose, mMRS + Arabinose, and mMRS + Xylose refers to the mean of the growth rates (OD600 $\mu_{max}$ ($h^{-1}$)). Cellulolytic activity refers to the mean of the DNS-based assay (mg $mL^{-1}$ min).

**Author Contributions:** Conceptualization, D.R.N. and O.M.; methodology, D.R.N. and O.M.; formal analysis, D.R.N.; writing—original draft preparation, D.R.N.; writing—review and editing, O.M. and M.C.; supervision, A.T.; project administration, O.M.; funding acquisition, O.M. All authors have read and agreed to the published version of the manuscript.

**Funding:** This research was funded by TEAGASC (project ref. MDBY0402). D. Román Naranjo is supported by a Teagasc Walsh Scholarship (ref. 2018036).

**Institutional Review Board Statement:** Not applicable.

**Informed Consent Statement:** Not applicable.

**Data Availability Statement:** Not applicable.

**Acknowledgments:** The authors would like to thank Peter Myintzaw (Munster Technological University) for assistance with Growthcurver.

**Conflicts of Interest:** The authors declare no conflict of interest.

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
