# Peer review of "Evaluation of Environmental Lactococcus lactis Strains Reveals Their Potential for Biotransformation of Lignocellulosic Feedstocks"

_2673-8007, doi:10.3390/applmicrobiol2040061_

Round 1

Reviewer 1 Report

This article focuses on the potential of several wild-type Lactobacillus lactis (L. lactis) isolated from natural environment to metabolize lignocellulosic breakdown products. The author has done a lot of work, but the experiment is not innovative enough and there are still many problems:

1. An extra comma on lines 15-16.

2. In lines 81-86, the highlights of the biotransformation of lignocellulose by naturally isolated strains compared with the fermentation of lignocellulose by genetically engineered bacteria should be introduced.

3. In lines 86-90, what is the difference between the genetic and physiological perspectives? What are the highlights from the physiological perspective?

4. There are too many tables in this article, please try other types of statistical charts.

5. The content of the experiment was too simple, and only the ability of pentose fermentation and cellulose hydrolysis of the strain was tested, which was not innovative enough.

Author Response

  1. An extra comma on lines 15-16.

The extra comma has been removed from lines 15-16.

  1. In lines 81-86, the highlights of the biotransformation of lignocellulose by naturally isolated strains compared with the fermentation of lignocellulose by genetically engineered bacteria should be introduced.

The main highlights of xylose fermentation by genetically engineered and non-engineered L. lactis strains have been highlighted:

Lines 87-91: Even though the mutant L. lactis strain IL 1403 produced 50.1 g L-1 of L-lactic acid from xylose [22], the wild strain L. lactis IO-1 produced a maximum concentration of 33.26 g L-1 of lactic acid from xylose [27]; this shows the potential of non-engineered L. lactis to metabolize this complex carbon source and to generate compounds of interest.

  1. In lines 86-90, what is the difference between the genetic and physiological perspectives? What are the highlights from the physiological perspective?

An example has been added about the main differences between genetic and physiological approaches

Lines 93-97: The main difference between the physiological and the genomic perspective is the lack of correlation between the presence of specific genes and the phenotypic activity assigned to those genes, which happens sometimes, mainly due to gene inactivation [28]. For example, the presence of a xylose utilisation gene cluster in strain L. lactis subsp. lactis IL-1403 did not correlate with the ability of this strain to utilise xylose

  1. There are too many tables in this article, please try other types of statistical charts.

Table 3 has now been changed to Figure 3 (bar chart)

Line 263: Figure 3: Cellulolytic activity (mg ml-1 min-1) in the quantitative DNS assay for all the environmental L. lactis isolates that were positive on CMC agar. The values represent the average of triplicates (mean) and the error bars the standard deviation (SD).

In addition, Table 2 was simplified by suppressing the third column (isolation source), which were redundant with Table 1

  1. The content of the experiment was too simple, and only the ability of pentose fermentation and cellulose hydrolysis of the strain was tested, which was not innovative enough.

We believe this work is innovative because it explores and expands the untapped metabolic capabilities of environmental L. lactis strains and it advances the field of lignocellulose biotransformation by providing another set of strains (6 strains) capable of both metabolizing the sugars derived from its breakdown and also cellulose. This could potentially contribute to a circular economy and help mitigate the issues of food waste and climate change, due to the enormous potential of lignocellulose as an inexpensive, abundant and renewable energy source.

Reviewer 2 Report

see attached document

Author Response

Answers:

  • The growth media is modified MRS (mMRS) made from scratch, which does not contain any sugar source. This experiment measured the differences of the strains when growing on different sugars. The reference here is comparison with the growth on the glucose sugar. In order to clarify this, MRS term has been changed by mMRS throughout the manuscript and supplementary files, indicating modified MRS media. Moreover, this screening approach has been previously used to successfully identify lactic acid bacteria with potential to metabolize those pentoses (Boguta et al., 2014).
  • The concerns about the accuracy of the DNS assay have been expressed in the Discussion section of the article.

Lines 374-376: Nonetheless, we understand the recent concerns about the use and the accuracy of DNS assay for the reducing sugars since other carbonyl groups could potentially react with the DNS leading to incorrect yields of reducing sugars [51].

Specific comments:

1.- Line 17.... Eleven of the 21 L. lactis isolates were able to ferment pentose sugars … Lines 21-22….Eleven environmental  L. lactis strains were found to  have the potential to ferment pentose sugars  ….

I consider that the same result is reported twice in the Abstract section. Please rewrite this

Lines 21-22 have now been eliminated in order to avoid repetition

2.- Line 66...   resistant to microbial breakdown to fermentable sugars  [11,12]. ….

Lignin cannot be breakdown to sugars,  as the authors state, because lignin does not contain sugars in its chemical structure. Tentatively, lignin could be used as carbon source to produce sugars. Please rewrite  this sentence.

The sentence has been rewritten

Line 66: Lignin is the second most abundant renewable polymer after cellulose. It is also the most structurally complex lignocellulose component (highly cross-linked phenylpropanoid units), which makes it resistant to be used by microorganisms to produce monomers [12,13].

3.- The manuscript and supplementary material contain units and names such as:   hours..., 3,5-Dinitrosalicylic acid,... L. lactis,...  Lactiplantibacillus pentosus, Levilactobacillus brevis and L actobacillus plantarum... ( h-1)) 

Units should be indicated correctly (e.g. h instead of hour or hours; h-1 instead of h-1  ).

Chemical names should be written in lower case unless they are placed immediately after a dot (e.g. 3,5-dinitrosalicylic instead of 3,5-Dinitrosalicylic).

Strain names should be written in italics (e.g. L. lactis, Lactiplantibacillus pentosus, Levilactobacillus brevis and L actobacillus plantarum instead of L. lactis, Lactiplantibacillus pentosus, Levilactobacillus brevis and L actobacillus plantarum

This has all now been corrected.

4.- Reference #14

Please, add journal name.

Journal name has been added to reference 14.

Reviewer 3 Report

It’s an interesting paper to study a species commonly used in food production, L. lactis, isolated from a variety of grass and vegetables, to metabolize the sugars derived from lignocellulose. Both cellulolytic activity and pentose sugars metabolizing ability were detected in six strains out of 21 evaluated strains. A proof reading by a native English speaker is recommended to improve both language and organization quality. This paper is fairly complete and should be published. It can be improved with the following suggestions:

Line 43-44,  what is the remaing 40% composition in plant biomass?

Line 48-51, please indicate  biological treatment has some disadvantages such as slow process ?

Line 78-81,  why do you comare in terms of lipolytic activity, not sugar metabolizing ability?

Line 220-223, do isolated strains have the preference to metabolize arabinose or xylose?

Line 254-269, have you done any genome sequencing for the isolated strains?

Do you find any inhibition effect in the lignocellose hydrolysate to the isolated strains?

Author Response

It’s an interesting paper to study a species commonly used in food production, L. lactis, isolated from a variety of grass and vegetables, to metabolize the sugars derived from lignocellulose. Both cellulolytic activity and pentose sugars metabolizing ability were detected in six strains out of 21 evaluated strains. A proof reading by a native English speaker is recommended to improve both language and organization quality. This paper is fairly complete and should be published. It can be improved with the following suggestions:

Two of the authors of this paper are native English speakers.

Line 43-44, what is the remaing 40% composition in plant biomass?

The sentence has now been amended

Lines 42-44: Lignocellulose, a polymer which is the most abundant biomaterial on Earth and one of the most abundant renewable feedstocks, comprises on average 23% lignin, 40% cellulose, and 33% hemicellulose by dry weight.

Line 48-51, please indicate biological treatment has some disadvantages such as slow process ?

This has now been included

Lines 50-51: Nonetheless, biological treatments also have some disadvantages such as low hydrolysis rate [7].

Line 78-81, why do you compare in terms of lipolytic activity, not sugar metabolizing ability?

Our purpose with this comparison was to show the more diverse metabolic capabilities of these strains, which are the premises of exploring other metabolic capabilities such as lignocellulose metabolization.

Line 220-223, do isolated strains have the preference to metabolize arabinose or xylose?

This has now been included

Lines 234-235: Most strains had higher growth rates on arabinose rather than xylose, and only two strains had higher growth rates in xylose compared to arabinose: DPC6765 and DPC6730.

Line 254-269, have you done any genome sequencing for the isolated strains?

Only 6 L. lactis isolates were previously sequenced.

As indicated in the Discussion in lines 323-328: A preliminary study was carried out to associate specific genes to the phenotypic activities. However, it did not clearly show a genotype-phenotype correlation with regard to metabolism of these sugars.

Do you find any inhibition effect in the lignocellose hydrolysate to the isolated strains?

Inhibition effects were not measured in this study.

Round 2

Reviewer 2 Report

The authors have carefully addressed most of the in-depth revisions suggested by this reviewer and provided convincing corrections and kind replies. The paper is really improved and can be accepted.